# Unravelling an oxygen-mediated reductive quenching pathway for photopolymerisation under long wavelengths

Chenyu Wu [1,2,3], Kenward Jung[1,2], Yongtao Ma[3], Wenjian Liu [3✉] & Cyrille Boyer [1,2✉]

Photomediated-reversible-deactivation radical polymerisation (photo-RDRP) has a limited scope of available photocatalysts (PCs) due to multiple stringent requirements for PC properties, limiting options for performing efficient polymerisations under long wavelengths. Here we report an oxygen-mediated reductive quenching pathway (O-RQP) for photoinduced electron transfer reversible addition-fragmentation chain transfer (PET-RAFT) polymerisation. The highly efficient polymerisations that are performed in the presence of ambient air enable an expanded scope of available PCs covering a much-broadened absorption spectrum, where the oxygen tolerance of PET-RAFT allows high-quality polymerisation by preventing the existence of $O_2$ in large amounts and efficient O-RQP is permitted due to its requirement for only catalytic amounts of $O_2$. Initially, four different porphyrin dyes are investigated for their ability to catalyse PET-RAFT polymerisation via an oxidative quenching pathway (OQP), reductive quenching pathway (RQP) and O-RQP. Thermodynamic studies with the aid of (time-dependent) density functional theory calculations in combination with experimental studies, enable the identification of the thermodynamic constraints within the OQP, RQP and O-RQP frameworks. This knowledge enables the identification of four phthalocyanine photocatalysts, that were previously thought to be inert for PET-RAFT, to be successfully used for photopolymerisations via O-RQP. Well-controlled polymerisations displaying excellent livingness are performed at wavelengths in the red to near-infrared regions. The existence of this third pathway O-RQP provides an attractive pathway to further expand the scope of photocatalysts compatible with the PET-RAFT process and facile access to photopolymerisations under long wavelengths.

[1] Centre for Advanced Macromolecular Design (CAMD), School of Chemical Engineering, University of New South Wales Sydney, NSW 2052, Australia. [2] Australian Centre for NanoMedicine, University of New South Wales Sydney, NSW 2052, Australia. [3] Qingdao Institute for Theoretical and Computational Sciences, Shandong University, Qingdao 266237, China. ✉email: liuwj@sdu.edu.cn; cboyer@unsw.edu.au

Photoredox catalysis has proven to be a powerful strategy in organic transformations[1] and macromolecular syntheses[2–9], especially for reversible-deactivation radical polymerisation (RDRP)[10–12]. In a photo-RDRP system[5,8,12–15], a specific quenching pathway describes how the photoexcitation energy is converted to chemically activate the polymerisation through a series of single electron transfer (SET) or energy transfer reactions. Established quenching pathways in photoredox catalysis include both the oxidative quenching pathway (OQP, Fig. 1a left), which involves oxidation of an excited photocatalyst (PC) and the reductive quenching pathway (RQP, Fig. 1a, middle), which involves reduction of an excited PC[1]. The initiating species of the photo-RDRP process can thus be activated by the excited PC via the OQP or by the PC anion (produced by reducing the excited PC) via the RQP.

Despite the successful implementation of these pathways to regulate photo-RDRP, the scope of efficient PC candidates is however very much limited (Fig. 1a). This is primarily due to the stringent requirements for an efficient PC to serve as a sufficiently strong excited-state reductant in OQP or excited-state oxidant in RQP[16]. In the context of photo-RDRP, increasing catalyst versatility is vital for its successful implementation in various application scenarios and media. Furthermore, in stark contrast to thermal reactions which proceed at a single elevated temperature, the narrow absorptions of PCs extending into the visible and near-infrared (NIR) ranges can be exploited for multiple concurrent, potentially orthogonal, reactions in the same volume[17–20]. In particular, NIR light provides promoted light penetration and reduced side reactions as a benign light source[21–23]. Previously, we discovered that a metalloporphyrin, zinc octaethyltetraphenyl porphyrin, could not facilitate polymer synthesis via the established OQP or RQP mechanisms and only in the presence of atmospheric oxygen did the polymerisation proceed, with inert gases providing a means to temporarily pause the process. Critically, polymerisations were not observed in the absence of RAFT agent; this observation, in conjunction with the polymerisation proceeding at accelerated rates in the presence of pure oxygen, led to the hypothesis that oxygen (O$_2$) could be playing the role of a mediator to specifically activate the RAFT agent.

In this work, experimental and computational studies are combined to clarify and validate the underlying mechanism of this pathway, which we have designated as O$_2$-mediated RQP (O-RQP, Fig. 1a, right). From an initial group of four different PCs, varied selectivity of the quenching pathways is identified and used as the basis for discerning the thermodynamic requirements of different pathway in silico. This knowledge facilitates the identification of four other O-RQP compatible catalysts. The mechanistic pathway established herein provides an oxygen-tolerant RDRP process that is particularly efficient with PCs capable of absorbing low-energy photons, with wavelengths ranging from orange to the NIR region without activating the established OQP and RQP pathways[1].

## Results

**Experimental screening of PCs.** Despite significant effort devoted to the discovery of PCs for PET-RAFT polymerisation via the OQP or RQP, the scope of efficient PCs available is still very limited, especially for candidates with longer-wavelength light absorption. This is primarily due to strict thermodynamic constraints of PCs in the OQP and RQP (Fig. 1). In the OQP, OQP-I is a process where $^3$PC* absorbs energy and donates an electron from its upper singly occupied molecular orbital (upper-SOMO). Consequently, a $^3$PC* with lower-lying upper-SOMO is less likely to activate OQP-I (requires more energy to donate an electron) and can thus be ineffective for the OQP. Conversely, in the RQP, RQP-I is a process where $^3$PC* accepts an electron (releasing energy) to its lower-SOMO leading to RQP-II where the PC$^{•-}$ donates that electron (absorbing energy) from its only SOMO. Hence, either a higher-lying lower-SOMO of $^3$PC* (releasing less

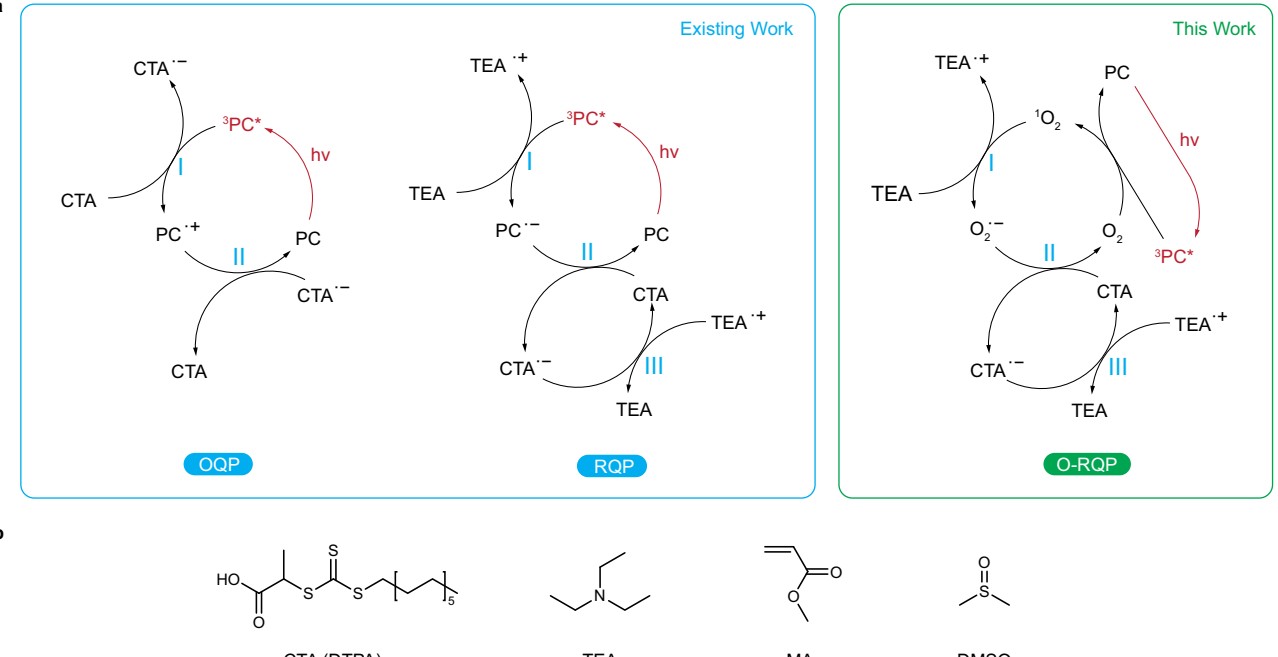

**Fig. 1 Model PET-RAFT polymerisation and mechanisms. a** Catalytic cycles for the model PET-RAFT polymerisation via OQP (left), RQP (middle) and O-RQP (right). **b** Molecular structures for the chain-transfer agent, 2-(dodecylthiocarbonothioylthio)propionic acid (CTA, DTPA), co-catalyst, triethylamine (TEA), monomer, methyl acrylate (MA) and solvent dimethyl sulfoxide (DMSO) for the model PET-RAFT polymerisation. OQP oxidative quenching pathway, RQP reductive quenching pathway, O-RQP oxygen-mediated reductive quenching pathway.

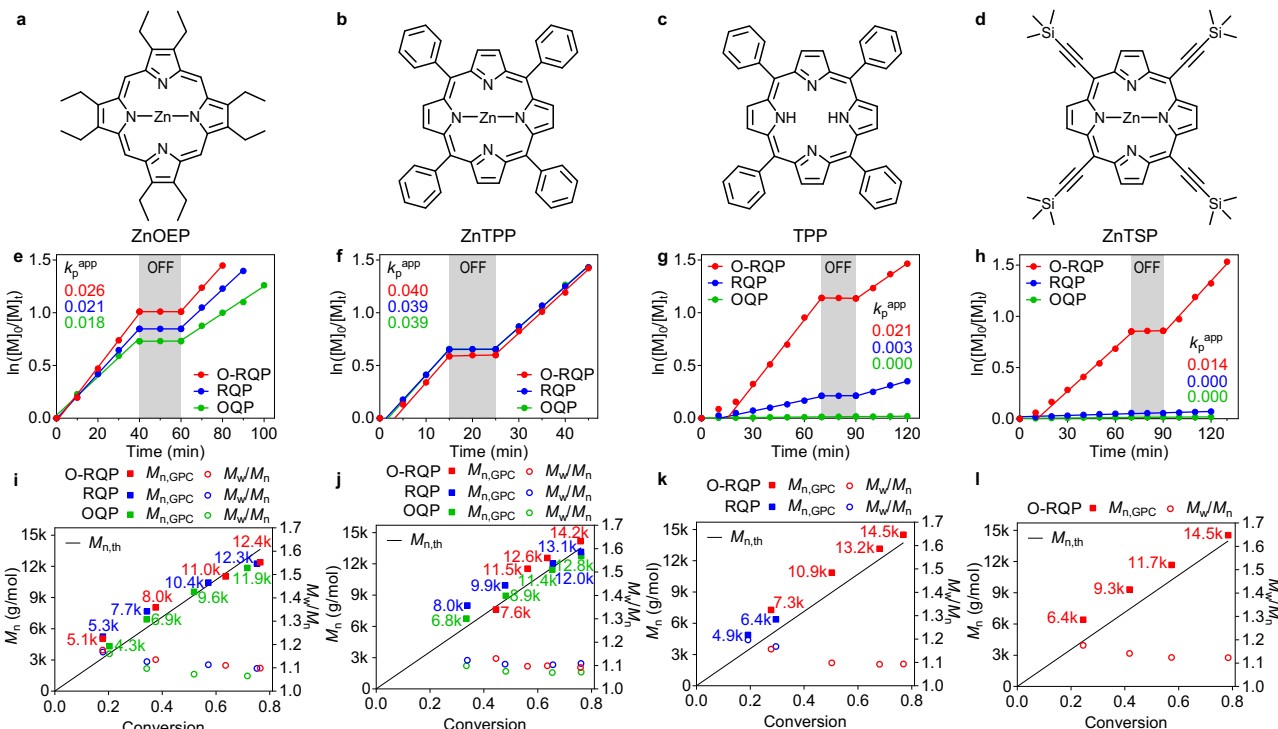

**Fig. 2 Kinetic studies of PCs via different quenching pathways. a–d** Molecular structures of ZnOEP (**a**), ZnTPP (**b**), TPP (**c**) and ZnTSP (**d**). **e–h** Plot of ln ($[M]_0/[M]_t$) versus time revealing $k_p^{app}$ and temporal control for model PET-RAFT polymerisation via OQP (green), RQP (blue) and O-RQP (red) catalysed by ZnOEP (**e**), ZnTPP (**f**), TPP (**g**) and ZnTSP (**h**) respectively. **i–l** $M_n$ (solid square) and $M_w/M_n$ (hollow circle) versus monomer conversion analysed by gel permeation chromatography from aliquots taken at different time intervals during model PET-RAFT polymerisation via OQP (green), RQP (blue) and O-RQP (red) catalysed by ZnOEP (**i**), ZnTPP (**j**), TPP (**k**) and ZnTSP (**l**), respectively. The associated molecular weight distributions are provided in Supplementary Figs. 2–10. $k_p^{app}$ apparent propagation rate of polymerisation, the number of which is denoted aside the corresponding polymerisation slope with the same colour. Average molecular weights $M_n$ were denoted aside the corresponding points in **i–l** with the same colours.

energy to accept an electron) or lower-lying SOMO of PC•− (requiring more energy to donate an electron) can increase the overall energy requirements, decreasing the efficiency to which $^3PC^*$ can facilitate the RQP. Indeed, these complex thermodynamic constrains have excluded a great number of dyes from being effective or efficient for PET-RAFT polymerisation. These constraints could also be responsible for many selectivities of certain PCs with respect to specific RAFT agents[24,25].

The discovery of O-RQP originates from our recent finding that suggested singlet oxygen ($^1O_2$) generated by $^3PC^*$ can react with triethylamine (TEA) to form superoxide ($O_2^{•−}$), which can further interact with the $^3PC^*$ and activate the chain-transfer agent (CTA)[26]. This deviated from the known RQP pathway in that no polymerisation was observed in the absence of oxygen, suggesting the existence of an alternative activation pathway. In following re-evaluation, we discovered that the interaction between $O_2^{•−}$ and $^3PC^*$ is not necessary to activate the CTA, as $O_2^{•−}$ is thermodynamically capable of directly activating trithiocarbonates (vide infra). With the onus of activation lying with superoxide, this mechanism has the potential to be generalised and expanded to encompass a much broader range of chromophores. To investigate this potential, we first compared a selection of four chromophores, two that were known to be compatible with PET-RAFT and two other candidates. First, zinc (II) 5,10,15,20-(tetraphenyl)porphyrin (ZnTPP, Fig. 2b), which has exhibited excellent efficiency towards PET-RAFT at extended wavelengths, was selected as the baseline in terms of thermodynamic characteristics associated with effective OQP catalysis. Although possessing a similar structure to that of ZnTPP, the non-metallated 5,10,15,20-(tetraphenyl)porphyrin (TPP, Fig. 2c)

has exhibited significantly worse ability to regulate PET-RAFT via the OQP. Additionally, its capability to mediate PET-RAFT via the RQP has yet to be investigated and therefore provided an interesting proposition for this study. From these two known PCs, we further selected zinc(II) 2,3,7,8,12,13,17,18-octaethylporphyrin (ZnOEP, Fig. 2a) and zinc(II) 5,10,15,20-(tetra-trimethylsilylethynyl)porphyrin (ZnTSP, Fig. 2d); owing to the lack of electron-withdrawing phenyl groups, the former is likely to be a more reductive PC in comparison to ZnTPP. In contrast, the ethynyl substituents of ZnTSP leads to an expanded macrocycle and hence more stabilised (in terms of π* orbitals) species that requires lower excitation energy at the expense of weaker excited-state redox capabilities. Detailed discussions with respect to the structure–property relationships are presented in Supplementary Note 5. Although sharing structural similarities, the four PCs were predicted to present varying degrees of photocatalytic capabilities and therefore enabled the isolation of the O-RQP mechanism for more detailed investigation.

A model PET-RAFT system was used for consistency in all cases, with 2-(dodecylthiocarbonothioylthio)propionic acid (DTPA) as the chain-transfer agent (CTA), methyl acrylate (MA) as the monomer and dimethyl sulfoxide (DMSO) as the solvent (Fig. 1b) for the OQP, at a fixed molar ratio of [CTA]: [monomer]:[PC] = 200:1:0.01 and monomer to solvent 1:1 v/v (details see Supplementary Note 3). The selection of DTPA and MA as the model system was not only in consideration that they are commonly used in kinetic studies but also because the thermodynamic properties of DTPA before and after one or more MA additions are essentially the same (see Supplementary Note 3 and Supplementary Fig. 1), which is ideal for thermodynamic

studies and theoretical evaluation. To enable the RQP, the model co-catalyst triethylamine (TEA, Fig. 1b) was added at 0.5 equivalence to the CTA. To enable an O-RQP, the same recipe used for RQP was sealed without deoxygenation (in the presence of air). In contrast, polymerisations via an OQP or the RQP were deoxygenated by purging with nitrogen for 10 min prior to light irradiation. Nominal wavelengths of the light source were chosen according to the red-most absorption peak (the lower energy Q-band) of the porphyrin-based molecule, i.e. 565 nm yellow light for ZnOEP, 590 nm orange light for ZnTPP, 625 nm red light for TPP, and 660 nm deep-red light for ZnTSP at 10 mW/cm$^2$ using light-emitting diodes (LEDs, more details for consideration of selecting irradiation sources see Supplementary Note 3).

Utilising this standardised approach, the polymerisation kinetics mediated by each of the selected catalysts were investigated to ascertain differences in capabilities with respect to the three catalytic pathways (Fig. 2). Survey of the pseudo-first-order kinetic plots for each of the PCs (Fig. 2e–h) provides interesting insights into the varied capabilities of each species. Those possessing strong reducing capabilities, ZnOEP (Fig. 2e) and ZnTPP (Fig. 2f) were observed to be effective mediators of all three reaction conditions. In contrast, TPP showed limited ability to mediate PET-RAFT via the RQP and inability to facilitate the OQP in the investigated time frame. ZnTSP on the other hand was found to be effective for the oxygen-mediated pathway (O-RQP) only. In fact, with all four species demonstrating robust performance via this pathway, it leads to the suggestion that O-RQP may indeed be the "path of least resistance". The predominant reason for the inertness of ZnTSP in OQP and RQP is the four electron-withdrawing ethynyl groups which decreases the energy of the lowest unoccupied molecular orbital (LUMO; Supplementary Fig. 34, l) compared to the other PCs. As the lower-lying LUMO corresponds to a lower-lying upper-SOMO of the excited-state PC ($^3$PC*), the OQP is made less favourable. Additionally, the lower-lying SOMO of ZnTSP$^{•-}$ makes it less favourable for RQP and consequently it displayed the lowest activity among the four PCs (i.e., inert for OQP and RQP). Irrespective of the pathway in which the polymerisations proceeded, clear evidence of temporal control was demonstrated by the ability to temporarily pause the process with full recovery of the rate upon switching the light back on. Analysis by gel permeation chromatography revealed narrow molecular weights that clearly shifted towards higher molecular weight with increasing irradiation time for all cases (Supplementary Figs. 2–10; molecular weight dispersity $M_w/M_n$ ~1.1, Fig. 2i–l). Living characteristics of a linear first-order kinetic plot (Fig. 2e–h) and linear increases in molecular weight ($M_n$) with increasing conversion (Fig. 2i–l) were further supported by nuclear magnetic resonance (NMR) spectroscopy (Supplementary Fig. 16) and matrix-assisted laser desorption/ionisation-time of flight (MALDI-TOF) mass spectroscopy (Supplementary Fig. 18), which confirmed high end-group fidelity of the synthesised polymers. Notably, no noticeable side effects stemming from the presence of $O_2$ on the synthesised polymer were observed, consistent with our previous observations[27–29]. Chain extension experiments for all synthesised polymers were performed with dimethylacrylamide (DMA) to produce block copolymers; complete shifts of molecular weight distribution (MWD) towards higher molecular weights in conjunction with narrow molecular distributions ($M_w/M_n$ ~1.1, Supplementary Figs. 19–27), lend further support to the high retention of end groups in all cases. O-RQP polymerisations performed in various solvents and for different monomers also proved to be successful, demonstrating the generality of the approach (Supplementary Fig. 41 and Supplementary Table 17). Details are presented in Supplementary Note 11.

**Theoretical investigation of PCs.** The use of redox potentials to explain the activation of initiators by a PC for photo-RDRP is a reasonable approach when the electrochemical properties of the investigated PCs are to be qualitatively compared in a single photo-RDRP system with a given initiator[29–32]. However, when examining each catalytic mechanism in greater detail, such as the thermodynamic interplay between PCs and different quenching pathways, the above methods fall short of the required quantitative accuracy. In this work, where the thermodynamic favourability of a PC in activating different quenching pathways is of interest, the calculation of the change in Gibbs free energy ($\Delta G$) for each SET event, in conjunction with a theoretically derived activation threshold, provides a clearer picture with regard to the viability of the various SET events.

Each of the catalytic pathways are composed of steps involving both excited-state and ground-state species. Based on Marcus theory, SET reactions between excited-state species and ground-state species can usually be classified into two major types. The SET reaction between an excited-state reactant and a ground-state reactant (ES–GS SET, i.e., PET) is irreversible. On the other hand, an SET reaction between two ground-state species (GS–GS SET) is a reversible process[33]. For more details see Supplementary Note 8. In the case of GS–GS SET, we define $\Delta G$ as the Gibbs free energy change of an SET process (describing its favourability at a specific temperature, i.e. at room temperature here) and $\Phi_{SET,GS–GS}$ as the quantum yield of a GS–GS SET process (expressed as Supplementary Eq. 1 in Supplementary Note 8 describing how likely the reversible GS–GS SET ends up with products after the SET event). By collectively solving equations for the $\Phi_{SET,GS–GS}$ expression (Supplementary Eq. 1), the equilibrium constant $K$ expression[34–36] (Supplementary Eq. 2), the Nernst equation at equilibrium[37,38] (Supplementary Eq. 4), we derived the relation between $\Phi_{SET,GS–GS}$ and $\Delta G$ (Supplementary Eq. 5) and correspondingly derived the thermodynamic thresholds for GS–GS SET. This leads to the quantum yield of a GS–GS SET process ($\Phi_{ET,GS–GS}$) being equal to 50% at $\Delta G = 0$ kcal/mol, 1.45% at $\Delta G = 5$ kcal/mol, and 0.05% at $\Delta G = 9$ kcal/mol. Utilising the assumption that a quantum yield of less than 0.05% ($\Phi_{ET,GS–GS} < 0.05\%$) is inadequate to facilitate the generation of sufficient quantities of reactive species for the next step, provides us with a prohibitive threshold ($\Delta G > 9$ kcal/mol) wherein the reaction is deemed unlikely to proceed. On the other hand, when $\Delta G < 5$ kcal/mol, $\Phi_{ET,GS–GS} > 1.45\%$, and is considered to afford effective activation of the reactive step (activation threshold = $\Delta G < 5$ kcal/mol). For more details see Supplementary Note 8.

In the case of PET (ES–GS SET), because it is known to be irreversible[33], only the rate of the PET process $k_{PET}$ is the limiting factor for efficiency. Indeed, $k_{PET}$ of a specific PET process must be high enough for PET to occur within the excited-state lifetime of the photo-excited reactant, otherwise the PET process will not proceed. As Marcus theory clearly suggests that thermodynamics controls the kinetics of SET processes[39,40] (based on the Arrhenius[41] and Eyring[42,43] equations), there must be thermodynamic thresholds for the PET process. To quantify these thresholds, we introduced the Eyring equation[42,43] and established the relationship between $k_{PET}$ and $\Delta G$ (Supplementary Eq. 6 in Supplementary Note 8). Because the $T_1$ lifetime of porphyrins are generally on the ms-scale, PET with a half-life $t_{(1/2)}$ (expressed as Supplementary Eq. 7) <10 ms (corresponding to $\Delta G < 15$ kcal/mol by solving Supplementary Eqs. 6 and 7) can be considered efficient at room temperature (i.e. a notable number of PET events can proceed within the $T_1$ lifetime), whereas PET with a $t_{(1/2)} > 100$ ms (corresponding to $\Delta G > 16$ kcal/mol) is assumed ineffective at room temperature. Thus, we derived $\Delta G < 15$ kcal/mol to be considered efficient (activation threshold) and

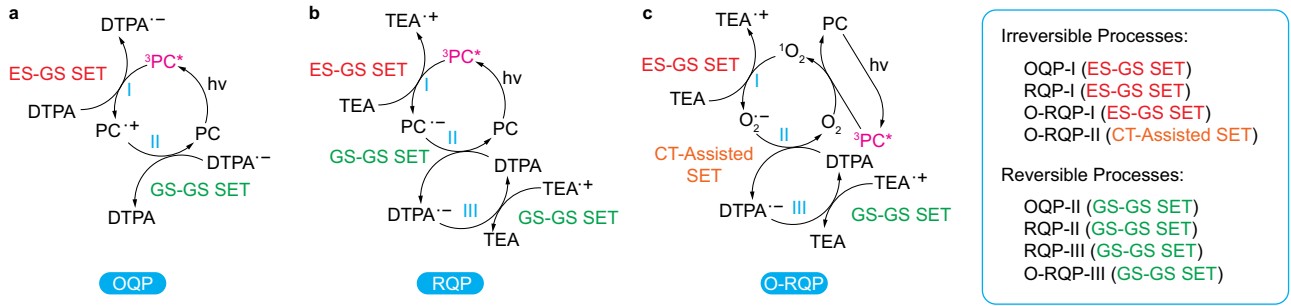

**Fig. 3 Categories of electron transfer steps. a–c** Nature of each electron transfer step in OQP (**a**), RQP (**b**) and O-RQP (**c**) catalytic cycles. OQP oxidative quenching pathway, RQP reductive quenching pathway, O-RQP oxygen-mediated reductive quenching pathway, ES–GS SET single electron transfer from an excited state to a ground state, GS–GS SET single electron transfer between two ground states, CT-assisted SET single electron transfer assisted by charge transfer in the donor–acceptor complex.

$\Delta G > 16$ kcal/mol to be considered ineffective (prohibition threshold). More details are discussed in Supplementary Note 8. With these boundary conditions established, we performed thermodynamics calculations for all four dye-catalysed photo-RDRP systems via OQP, RQP and O-RQP, respectively, to examine the theoretically derived thresholds. By comparing the thermodynamic properties of each specific SET processes in correlation with the experimentally observed selectivity, we confirmed the viability of the derived thermodynamic thresholds in practice (vide infra).

The OQP is perhaps the simplest quenching pathway in PET-RAFT, which only requires the presence of the PC to fulfil the catalytic cycle and activate the RAFT agent (Fig. 3a). This two-step pathway is composed of a PET process (OQP-I, Fig. 3a; possibility of energy transfer is discussed in Supplementary Note 7) that is followed by a GS–GS SET deactivation step (OQP-II, Fig. 3a). The two processes have distinct differences with regard to their thermodynamic viabilities; OQP-I, as a PET step, was deemed to be efficient when $\Delta G < 15$ kcal/mol and prohibited if $\Delta G$ exceeded 16 kcal/mol. On the other hand, OQP-II, which is a reversible GS–GS SET process composed of both forward and reverse SET reactions, can effectively proceed when $\Delta G$ is less than 5 kcal/mol and is suppressed when $\Delta G$ exceeds 9 kcal/mol. To clarify the thermodynamic viability of these steps with respect to the individual PCs, density functional theory (DFT) calculations were performed to derive the $\Delta G$ for both OQP-I ($\Delta G_I$, activation of RAFT) and OQP-II ($\Delta G_{II}$, deactivation of RAFT). As supported experimentally (Fig. 2e–h) and computationally (Fig. 4a–d), ZnOEP and ZnTPP are efficient PCs for OQP-I, while TPP and ZnTSP are unable to facilitate this crucial step (for consideration of exact kinetics see ref. [21]). Regardless of the viability of the deactivation step (OQP-II) for all PCs investigated herein, the inability to facilitate the OQP-I step highlights the major barrier precluding many potential candidates from application in the PET-RAFT process via oxidative quenching.

The RQP incorporates a co-catalyst, in this case, TEA, to facilitate reduction of the excited PC (ES–GS SET, RQP-I, Fig. 3b). A subsequent GS–GS SET between the PC radical anion (PC$^{\bullet-}$) and the RAFT agent initiates the polymerisation process (RQP-II, Fig. 3b). Finally, the active RAFT agent (CTA$^{\bullet-}$) is deactivated into the dormant species (CTA) via a GS–GS SET (RQP-III, Fig. 3b) with the TEA radical cation (TEA$^{\bullet+}$) that was originally formed during RQP-I. For all four PCs, it appears that the thermodynamic requirements for RQP-I ($^3$PC* reduction) and RQP-III (RAFT deactivation) are satisfied and it is the activation of the RAFT agent by the PC$^{\bullet-}$ (RQP-II), which determines the success of the reaction (Fig. 4e–h). TPP, while viable in this step ($\Delta G_{II} = 5 < 5.98 < 9$ kcal/mol), is significantly less efficient because of its relatively low $\Phi_{ET,GS-GS} = 0.64\%$. For

ZnTSP, this step is unviable, and polymerisation is therefore prohibited ($\Delta G_{II} = 12.81$ kcal/mol $> 9$ kcal/mol and $\Phi_{ET, GS-GS} = 0.00\%$), as confirmed experimentally.

By contrast, O-RQP is a less PC-dependant pathway. As shown both experimentally (Fig. 2e–h) and computationally (Fig. 4i–l), the O-RQP mechanism is a viable and effective pathway for all four PCs and leads to the suggestion that it is indeed less dependent on the redox capabilities of the PC. As revealed in the proposed mechanism (Fig. 3c), O-RQP-I, O-RQP-II and O-RQP-III are independent of the PC employed. Indeed, DFT calculations for O-RQP-I (the ES–GS SET process between $^1O_2$ and TEA to yield $O_2^{\bullet-}$ and TEA$^{\bullet+}$) exhibits $\Delta G_I = -1.14$ kcal/mol $< 15$ kcal/mol (highly favourable). As reported[26,44], $O_2^{\bullet-}$ was experimentally detected as the product of $^1O_2$-TEA reaction. To verify if the $^1O_2$-TEA reaction can substantially occur to yield $O_2^{\bullet-}$, we utilised ZnTSP (inert with OQP and RQP) as the photosensitiser to generate $^1O_2$ in the presence of the $^1O_2$ trapper, 9,10-dimethylanthracene (DMAn) to yield an endoperoxide, under 660 nm irradiation. The decreasing peak intensities of DMAn (402 nm) with irradiation time is indicative of $^1O_2$ generation, fulfilling the necessary first step of the proposed O-RQP mechanism (Supplementary Fig. 36, b). Addition of TEA into the same system ([PC]:[TEA] = 0.01:0.5, same as above) significantly retarded the consumption of DMAn. Further increase in the concentration of TEA ([PC]:[TEA] = 0.01:5) led to cessation of DMAn consumption (details see Supplementary Note 9). The suppression of endoperoxide formation with increasing TEA concentrations implied the competition for the same substrate (the TEA-$^1O_2$ reaction and the DMAn-$^1O_2$), with the formation of $O_2^{\bullet-}$ being evidently more efficient.

With the $O_2^{\bullet-}$-DTPA reaction seemingly the primary activation step we probed this reaction in greater detail. We performed DFT calculations on the bimolecular system and observed strong electrostatic affinity between DTPA and $O_2^{\bullet-}$ as well as significant intermolecular charge transfer (CT) from $O_2^{\bullet-}$ to DTPA, which is favourable for the $O_2^{\bullet-}$-DTPA SET to occur (Supplementary Fig. 40). On the other hand, the product DTPA$^{\bullet-}$-$O_2$ exhibits no electrostatic affinity nor CT character, which is favourable for rapid $O_2$ diffusion into the solution (Supplementary Fig. 40). Therefore, O-RQP-II can be considered a CT-assisted irreversible process that satisfies the activation threshold of 15 kcal/mol. Accordingly, we calculated $\Delta G$ of O-RQP-II to be 10.76 < 15 kcal/mol (Supplementary Note 10). With respect to $^1O_2$ photosensitisation, due to the complexities regarding the generation of $^1O_2$ (i.e. whether it exists as free $^1O_2$ or a (PC-$O_2$)* complex etc.)[45,46], we cannot perform further computational studies at present. Indeed, this aspect has remained in debate for decades and comprehensive ES dynamics is required to resolve the issue. With

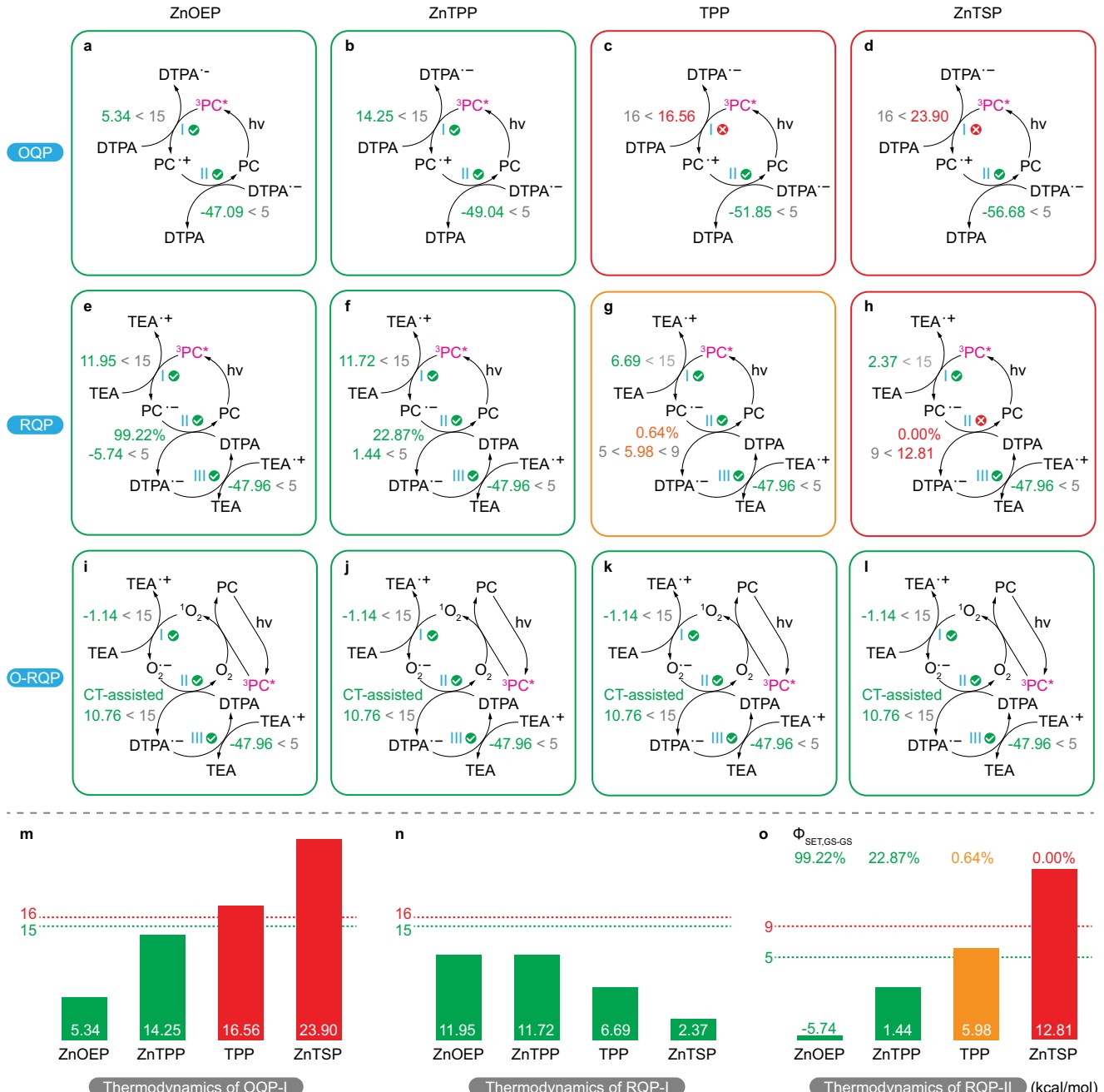

**Fig. 4 Schematic representation of thermodynamic viability of mechanistic steps. a–l** Mechanistic diagrams with $\Delta G$ of key steps denoted in comparison to corresponding thresholds for PET-RAFT polymerisation via OQP (**a–d**), RQP (**e–h**) and O-RQP (**i–l**), catalysed by ZnOEP (**a, e, i**), ZnTPP (**b, f, j**), TPP (**c, g, k**) and ZnTSP (**d, h, l**), respectively. Green is favourable, orange is less efficient and red indicates inertness. Coloured values are the $\Delta G$ of the corresponding process in kcal/mol, in conjunction with the threshold in grey (derived in the "Methods" section). Coloured percentage values are $\Phi_{SET,GS-GS}$ of the corresponding process. Bottom: Bar charts illustrating $\Delta G$ of the critical catalytic steps regarding the activation of OQP (OQP-I, **m**) and RQP (RQP-I, **n** and RQP-II, **o**) for ZnOEP, ZnTPP, TPP and ZnTSP on a comparative basis, with thermodynamic thresholds in kcal/mol (green dotted line: activation threshold; red dotted line: activation limit). $\Phi_{ET,GS-GS}$: quantum yield of the single electron transfer process between two ground states, defined by Supplementary Eq. 1. OQP oxidative quenching pathway, RQP reductive quenching pathway, O-RQP oxygen-mediated reductive quenching pathway. ES–GS SET single electron transfer from an excited state to a ground state, GS–GS SET single electron transfer between two ground states, CT-assisted SET single electron transfer assisted by charge transfer in the donor–acceptor complex.

the combination of theoretical and experimental results suggesting the thermodynamic favourability of $O_2^{•-}$ generation at room temperature, it appears that the predominant role of the PC in the O-RQP process is that of a photosensitiser to generate $^1O_2$.

As suggested by our previous contribution, the dual role of $O_2$ as a radical scavenger and a catalytic species, is a "paradox"[26].

Specifically, the propagating carbon radical in a polymerisation is subject to oxidation by ground-state $O_2$ leading to the formation of a peroxy radical[47] and degradation of the dormant polymer chain may also occur in the presence of ground-state $O_2$[48]. However, unlike conventional, thermally initiated RAFT polymerisations, the PET-RAFT systems possess excellent oxygen tolerance as reported in previous publications, where $O_2$ is rapidly

converted to $^1O_2$ upon irradiation and chemically quenched by the solvent DMSO, forming $DMSO_2$[24,27–29,49–51]. In this context, the interplay between the oxygen tolerance feature and the catalytic role of $O_2$ in PET-RAFT polymerisation via O-RQP mechanism lies in the amount of $O_2$; the oxygen tolerance feature facilitates high-quality polymerisation by lowering the $O_2$ concentration to the catalytic amounts required for efficient O-RQP.

To increase the readability of the supportive discussion of the O-RQP mechanism above, which is composed of a $^1O_2$ photosensitisation process, followed by O-RQP-I (electron transfer reaction between $^1O_2$ and TEA), O-RQP-II (activation of RAFT by superoxide $O_2^{\bullet-}$) and O-RQP-III (deactivation of RAFT) processes, we summarised the chain of evidence from both experimental and computational studies, supporting

literature and comparative studies as Supplementary Fig. 46 and Supplementary Note 14.

**Extension of O-RQP to phthalocyanines**. Confirmation of the reduced photocatalytic requirements for conducting O-RQP opens opportunities to expand the scope of PCs suitable for PET-RAFT. However, the production of reactive oxygen species (ROS), in the form of both $^1O_2$ and $O_2^{\bullet-}$, throughout the entire polymerisation process means that dyes susceptible to ROS-induced degradation may be unsuitable for this approach. Taking this limitation into consideration, we turned our attention to phthalocyanines. Phthalocyanines, as structural cousins to porphyrins, offer significantly easier preparative routes, making them the superior economical choice for industrial implementation[52–54]. Additionally, these species offer Q-bands that extend into the far-red to NIR regions with facile modification routes to facilitate further red-shifting; the application of phthalocyanines to PET-RAFT could greatly expand the range of serviceable wavelengths[23,55–57]. However, asides from utilisation in photodynamic therapy, the photocatalytic capabilities of phthalocyanines have been limited. The discovery of O-RQP presents more promise in that activation of the CTA is governed by the interaction of $O_2^{\bullet-}$ with the RAFT agent; the role of the phthalocyanine is simply to produce $^1O_2$ (refs. [55,58,59]).

Starting with the azaporphyrin zinc (II) 2,7,12,17-tetra-*tert*-butyl-5,1-,15,20-tetraaa-porphine (ZnTtBAzP), which possesses structural similarities with phthalocyanines but with unsaturated

**Table 1 Thermodynamic viabilities of PCs with respect to the critical steps in OQP and RQP.**

| OQP | OQP-I | RQP-I | RQP-II | |
|---|---|---|---|---|
| PC | $\Delta G_I$ (kcal/mol) | $\Delta G_I$ (kcal/mol) | $\Delta G_{II}$ (kcal/mol) | $\Phi_{SET,GS-GS}$ |
| ZnTtBAzP | 28.95 > 16 | 1.11 < 15 | 15.87 > 9 | 0.00% |
| ZnTtBPC | 23.48 > 16 | 9.81 < 15 | 12.47 > 9 | 0.00% |
| ZnOBOPC | 16.89 > 16 | 19.74 > 16 | 5 < 7.90 < 9 | 0.13% |
| ZnTtBNPC | 19.52 > 16 | 17.73 > 16 | 9.77 > 9 | 0.03% |

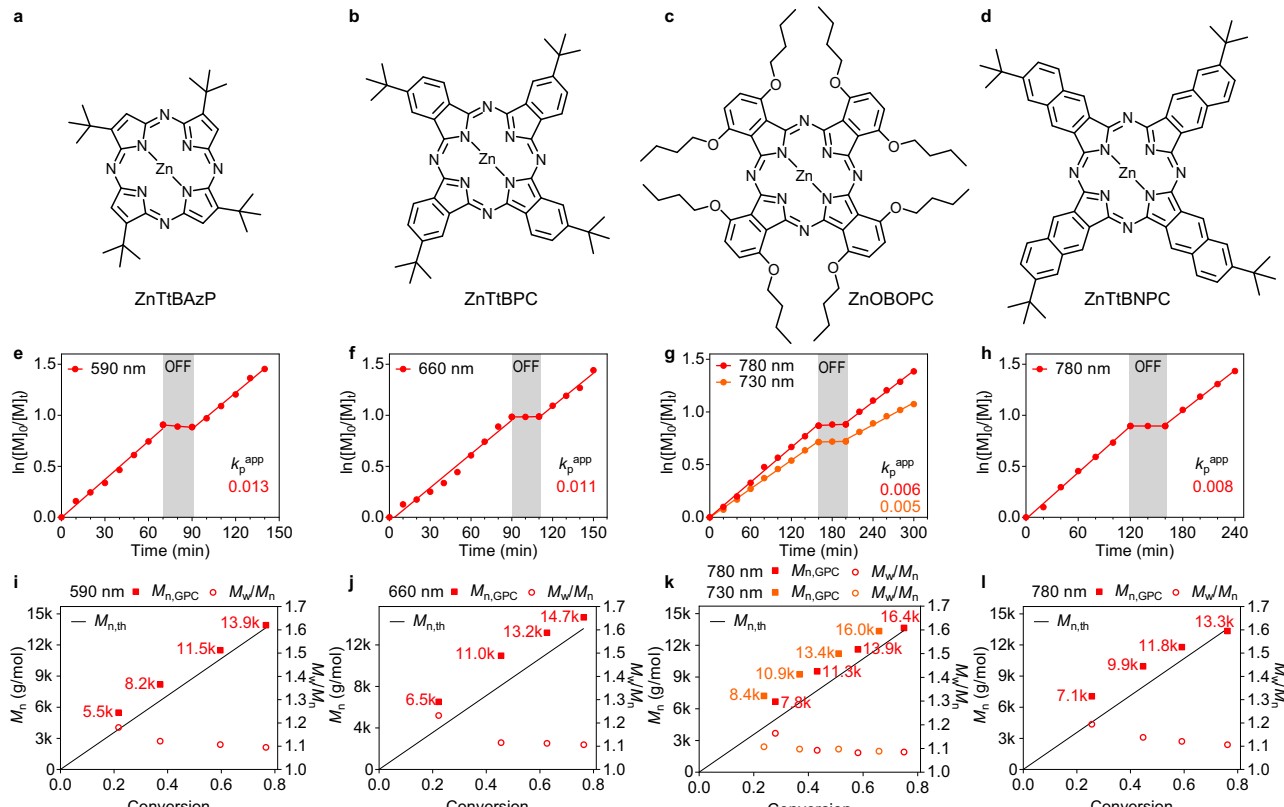

**Fig. 5 Kinetic studies of PCs via O-RQP. a–d** Molecular structure of ZnTtBAzP (**a**), ZnTtBPC (**b**), ZnOBOPC (**c**) and ZnTtBNPC (**d**). **e–h** Plot of $\ln([M]_0/[M]_t)$ versus time revealing $k_p^{app}$ and temporal control for model PET-RAFT polymerisation via O-RQP catalysed by ZnTtBAzP (**e**), ZnTtBPC (**f**), ZnOBOPC (**g**) and ZnTtBNPC (**h**), respectively. **i–l** $M_n$ (solid square) and $M_w/M_n$ (hollow circle) versus monomer conversion analysed by gel permeation chromatography from aliquots taken at different time intervals during model PET-RAFT polymerisation via O-RQP catalysed by ZnTtBAzP (**i**), ZnTtBPC (**j**), ZnOBOPC (**k**) and ZnTtBNPC (**l**), respectively. Associated molecular weight distributions were provided as Supplementary Figs. 11–15. $k_p^{app}$: apparent propagation rate of polymerisation, the number of which is denoted aside the corresponding polymerisation slope with the same colour. Average molecular weights $M_n$ were denoted aside the corresponding points in **i–l** with the same colours.

β-pyrrole positions, we sought PCs with increasingly red-shifted Q-bands. To this end, zinc (II) 2,9,16,23-tetra-*tert*-butyl-phthalocyanine (ZnTtBPC), zinc (II) 1,4,8,11,15,18,22,25-octabutoxy-phthalocyanine (ZnOBOPC) and zinc (II) 2,11,20,29-tetra-*tert*-butyl-2,3-napthalocyanine (ZnTtBNPC) were selected for their application in O-RQP PET-RAFT. For spectral details see Supplementary Note 5. Experimental screening (0% monomer conversion in 2 h) showed that these PCs were inert for both the OQP and RQP as expected. Thermodynamic studies (Table 1) further highlighted their difficulties with regard to the respective pathways. Interestingly, with all four dyes were calculated to be inert for RQP; ZnTtBAzP and ZnTtBPC were ineffective for RQP-II whereas ZnOBOPC was ineffective for RQP-I and ZnTtBNPC ineffective for both steps. More detailed discussions are presented in Supplementary Note 6.

Despite their inability to facilitate the OQP or RQP PET-RAFT polymerisations, these selected examples were highly effective in promoting the O-RQP process under irradiation wavelengths ranging from 590 to 780 nm (Fig. 5). In line with the results obtained for ZnTSP, the polymerisations performed using these dyes exhibited living controlled characteristics, such as a linear first-order plot, wherein the interruption to the irradiation caused negligible impact on the recovery of the polymerisation rate upon resumption. Additionally, molecular weights increased linearly with increasing conversion and were concomitant with narrowing of the MWDs (Supplementary Figs. 11–15). End-group retention was confirmed by chain extensions, which resulted in complete shift of the MWDs towards higher molecular weights (Supplementary Figs. 28–32).

Overall, through an approach combining experimental studies and computational investigation, the mechanism of an oxygen-mediated reductive quenching catalytic pathway (O-RQP) was presented, which eliminates sufficient $O_2$ to the point where radical polymerisation proceeds unencumbered while the remaining catalytic amounts of $O_2$ serve as a co-catalyst to provide facile polymerisation. Critically, (TD-)DFT calculations provided guidelines highlighting the thermodynamic favourability of a PC for a given catalytic pathway. Among these, O-RQP only necessitates the PC to be capable of generating singlet oxygen ($^1O_2$). To demonstrate the lowered requirements for O-RQP, we investigated four phthalocyanine derivatives, which have seen limited use in a photocatalytic context, in their ability to catalyse PET-RAFT polymerisation. While their implementation in OQP and RQP modes failed to yield any polymers, they were highly successful within the O-RQP framework. Exploiting the extended Q-bands of these phthalocyanines, well-controlled polymerisations exhibiting excellent livingness were performed in the presence of oxygen. The existence of the O-RQP mechanism provides an accessible avenue to increasing the scope of PCs that can enable better usage of a wider electromagnetic spectrum, especially long wavelengths in the NIR range.

## Methods

**General procedures for computation.** Geometry optimisations for PCs (and other species) were performed at the B3LYP-GD3BJ/6-311G* level of theory (pseudo-potential basis set LanL2TZ for zinc; SMD-DMSO solvation model), with the Gaussian09 D01 software package[60] installed in the Katana high-performance computing clusters at the University of New South Wales. The $T_1$ states were calculated by UDFT. Single-point energy calculations were further performed at the B3LYP-GD3BJ/6-311+G** level of theory for both ground states and $T_1$ states (UDFT). More details see Supplementary Note 13.

**General procedures for thermodynamic studies.** To evaluate the thermodynamics of each electron transfer catalytic process, the Gibbs free energy change (ΔG) was calculated as $\Delta G = G(P_1) + G(P_2) - G(R_1) - G(R_2)$, where $G(P_1)$ and $G(P_2)$ represent the Gibbs free energies ($G$) of products while $G(R_1)$ and $G(R_2)$ represent those of reactants. $G$ of each species was taken as the electronic energy as

common practice, since this is sufficiently accurate for thermodynamic studies of electron transfer events.

## Data availability

The authors declare that the data supporting the findings of this study are available within the paper and its Supplementary Information.

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

## Acknowledgements

C.B. acknowledges the Australian Research Council (ARC) for his Future Fellowship (FT120100096) and discovery project (DP180102540 and DP190100067). W.L. acknowledges National Natural Science Foundation of China (Grant Nos. 21973054 and 21833001).

## Author contributions

C.W.: writing the first draft; theoretical and experimental investigation; performing (TD-) DFT calculations. K.J.: scientific discussion; revision of the manuscript; and editing. Y.M.: scientific examination of thermodynamics. W.L.: scientific examination; advising and revision of all computational and theoretical discussions; supervision of (TD-)DFT calculations. C.B.: principle investigator; concept of the project; scientific discussion; supervision of experimental investigation and polymerisation applications; revision of the manuscript; editing.

## Competing interests

The authors declare no competing interests
