## [Peer Review File · Nature Communications]

REVIEWER COMMENTS

Reviewer #1 (Remarks to the Author):

Boyer and co-workers present a mechanism for PET-RAFT that uses oxygen as a catalytic species. The use of oxygen as a catalytic species allows new catalysts to mediate PET-RAFT as well as allows some catalysts to achieve higher performance. The role of oxygen in PET-RAFT has received considerable interest and this insight will be of significance to the field. This work appears to have been carefully performed and there is sufficient information in the main text and supporting information to enable reproducibility. This reviewer believes that this manuscript should be accepted for publication in Nature Communications as is but the authors may consider this advice. Figure 4 has a lot of information in it that may be difficult for readers to digest. Could this figure be re-made to be plotted on an energy scale relative to one another and the thermodynamic considerations?

Reviewer #2 (Remarks to the Author):

In this paper, the authors proposed a new mechanism for radical polymerization via oxygen-mediated reductive quenching pathway (O-RQP) under near IR irradiation, which is a nice expansion of the previous works by the same authors. The new plausible mechanism seems very interesting and, in fact, the authors have succeeded in controlling the polymerization using this catalyst. However, the reviewer thinks that it is premature to conclude the proposed mechanism, i.e., the photocatalyst activates an oxygen molecule into singlet oxygen that is further transformed into oxygen radical anion reacted with TEA and the oxygen radical anion finally causes the electron transfer to CTA. The process seems too complex and is slightly to the fact that (zinc) porphyrin induces photo oxidation in the presence of air, sometimes forming radical species, in may literature. In RAFT polymerization, the presence of a very small amount of radical species makes it possible to obtain a polymer having a narrow molecular weight distribution, which should not always be started from the radical cation of CTA.

The reviewer thinks that the authors should try once again to clarify the mechanism including the reaction between the porphyrin complex and oxygen as well as that between the photo activated oxidized porphyrin complex with CTA. Overall, the reviewer thinks it is premature to publish this work in Nat Commun.

Reviewer #3 (Remarks to the Author):

The manuscript by Cyrille and coworkers reported a well performed study on oxygen-mediated reductive quenching pathway for PET-RAFT. It seems that the study originated from their previous serendipitous observation that oxygen may play a catalytic role in PET-RAFT system, in sharp contrast to the common knowledge that oxygen has strong ability of chain transfer in radical polymerization. The authors screened four porphyrin photocatalysts (PC) in terms of their catalytic abilities for PET-RAFT in the presence and absence of amine and oxygen. By this screening, the selectivity of the PCs to three different catalyst pathways, i.e., OQP, RQP, and O-RQP, had been made clear. On this basis, the authors calculated the change in Gibbs free energy, using density functional theory (DFT), for each single electron transfer process in their proposed mechanism. The calculated thermodynamic thresholds agreed with experimental results. More importantly, the theoretical results indicate that O-RQP is a pathway with the lowest energy barrier. Based on this new knowledge of the role of oxygen in PET-RAFT and DFT calculation, the scope of PCs was successfully expanded to phthalocyanines, a family of organic dyes which was previously regarded to be not effective in PET-RAFT.

In my opinion, this is a remarkable progress, with deep inside mechanistic investigation, in the field of controlled polymerization which is of great interest for both academic and industry. I recommend to publish the work by Nature Communication on the basis of the following novelty of the paper. 1) A new mechanism, namely O-RQP, was proposed for the PET-RAFT in the presence of amine and oxygen. In this unique mechanism, a catalytic role of oxygen was revealed in photo-mediated reversible deactivation radical polymerization (photo-RDRP); 2) Technically,

thermodynamics of the electron transfer process was calculated using DFT, and discussed with respect to the molecular structure of the PC. This is in contrast to the commonly used correlation between redox potential and molecular structure; Thus the present work opens a paradigm shift in the research of photo-initiation; 3) Under the guidance of the proposed mechanism and thermodynamic study, the scope of PCs was successfully extended to phthalocyanines effective under longer wavelengths.

I am concerning the possible multiple roles of oxygen in radical polymerization. On one hand, oxygen builds a bridge between the excited state of PC and the ground state of the electron donor, TEA, and the RAFT agent, thus facilitating the SET process to generate radicals, as illustrated in the present work. On the other hand, oxygen is a well known scavenger to carbon-centered radicals, forming peroxide radical and, as such, retarding the rate of radical polymerization with very high chain transfer coefficient. The consumption of oxygen in trithiocarbonate mediated RAFT polymerization in the presence of tertiary amine and without deoxygenation process has been reported (Fu, Q.; et al Polym. Chem. 2017, 8, 1519). It is not clear how the molecular oxygen behave as both a catalyst and a retarder to the reaction. If the latter would be considered, the catalytic ability may turn out to be more effective to compensate the inhibition caused by O₂. Furthermore, it was reported that thiocarbonylthio compounds can be oxidized by the peroxide radical in the RAFT polymerization system (Li, C.; et al J. Polym. Sci.: Part A. Polym. Chem. 2011, 49, 1351 & Barner-Kowollik, C. Polym. Chem. 2010, 1, 634). Although they claimed that there was no noticeable side effects observed stemming from the presence of oxygen, the authors are encouraged to address these two important concerns by conciliating the results of the present work and those in the literature.

Minor revision: p.8 Fig. 3a: Is this for GS-GS or ES-GS? The indication in the figure is not consistent with the caption.

Comments made by Prof. Junpo He

Response point-by-point:

Reviewer #1

- Figure 4 has a lot of information in it that may be difficult for readers to digest. Could this figure be re-made to be plotted on an energy scale relative to one another and the thermodynamic considerations?

Answer:

Many thanks for the reviewer's attention on the readability of the figures. According to the reviewer's suggestion, we plotted the thermodynamic considerations of the critical processes on an energy scale with a comparative basis for different photocatalysts. This new part was added for readers' convenience right below the original **Figure 4**, while the original scheme was also kept considering that it is still the most straightforward and simplest way to show the detailed thermodynamic data of different catalytic processes against different photocatalysts.

The revised figure is as below and updated in the manuscript:

Figure R1 | Schematic representation of thermodynamic viability of mechanistic steps. Top: Mechanistic diagrams with ΔG of key steps denoted in comparison to corresponding thresholds for PET-RAFT polymerisation via OQP (first row), RQP (second row) and O-RQP (third row), catalysed by ZnOEP (first column), ZnTPP (second column), TPP (third column) and ZnTSP (fourth column) respectively. Green is favourable, orange is less efficient and red indicates inertness. Coloured values are the ΔG of the corresponding process in kcal/mol, in conjunction with the threshold in grey (derived in the Methods section). Coloured percentage values are $\Phi_{ET,GS-GS}$ of the corresponding process. Bottom: Bar charts illustrating ΔG of the critical catalytic steps regarding the activation of OQP (OQP-I, left) and RQP (RQP-I, middle and RQP-II, right) for ZnOEP, ZnTPP, TPP and ZnTSP on a comparative basis, with thermodynamic thresholds in kcal/mol (green dotted line: activation threshold; red dotted line: activation limit).

Reviewer #2

2. However, the reviewer thinks that it is premature to conclude the proposed mechanism, i.e., the photocatalyst activates an oxygen molecule into singlet oxygen that is further transformed into oxygen radical anion reacted with TEA and the oxygen radical anion finally causes the electron transfer to CTA. The process seems too complex and is slightly to the fact that (zinc) porphyrin induces photo oxidation in the presence of air, sometimes forming radical species, in many literature. In RAFT polymerization, the presence of a very small amount of radical species makes it possible to obtain a polymer having a narrow molecular weight distribution, which should not always be started from the radical cation of CTA.

Answer:

Firstly, we are very grateful that the reviewer paid much attention to the mechanistic aspect. The reviewer suggested that the proposed mechanism of O-RQP “seems too complex”. Indeed, in the initial stage of exploration in this project, we had also considered this mechanism for a long time and had similar concerns. Although we considered alternative mechanisms, none of them support our experimental evidence and computational studies in totality, leading to the presented conclusion. It appears that only the proposed O-RQP scheme can be fully supported in a highly self-consistent way by the experimental data, computational calculations and literature reports (summarised and discussed in detail below; see **Figure R2** below). These were further confirmed by comparative studies (see **Figure R2**, bottom row). On the other hand, despite visible-light photocatalysis is still in its infancy with very limited scope of quenching pathways, similar three-component catalytic cycles with comparable “complexity” to O-RQP have been reported in multiple researches. For example, König and coworkers reported a “sensitisation-initiated electron transfer” quenching pathway similar to O-RQP but replacing oxygen with pyrene,¹ whose mechanism was further revisited and proved by the same authors² and Moore and coworkers.³

The reviewer asserted that “(zinc) porphyrin induces photo oxidation in the presence of air, sometimes forming radical species”. The reviewer is absolutely correct in that photoexcited (zinc) porphyrin (i.e. TPP and ZnTPP) can generate radicals.^{4,5} Indeed, for the TPP/ZnTPP systems, the O-RQP mechanism cannot be decoupled from the OQP and RQP pathways (OQP and/or RQP is highly activated for ZnTPP/TPP under the “O-RQP condition”). The main point of including TPP/ZnTPP in the manuscript was to demonstrate that O-RQP (ZnTSP (660 nm) cannot activate either OQP or RQP but can still mediate polymerisation through the O-RQP mechanism) is a lower energy pathway accessible by existing/well-established photocatalysts and not limited to exotic examples. Consequently, this study addresses the challenge of mediating photochemistry at longer wavelengths via the classical OQP/RQP pathways. Noteworthy, it is indeed the drawbacks of ZnTPP/TPP (i.e.,

generating radicals upon photoexcitation) that brings about the initiative/urgency for us to replace ZnTPP and TPP with more stable photocatalysts. However, in PET-RAFT polymerization, ZnTPP and TPP are still the mainstream photocatalysts as there is a scarcity of stable alternatives capable of operating through the classical OQP and RQP with similar efficiency at long wavelengths (> 600 nm). One of the highlights of the O-RQP mechanism is that it expands the scope of photocatalysts for PET-RAFT polymerisation to other more inert families such as phthalocyanine derivatives,^{6,7} which has been shown to be inert for PET-RAFT via the OQP or RQP.⁸ When utilising the proposed O-RQP mechanism, phthalocyanine derivatives were successfully implemented for PET-RAFT polymerisation, circumventing the instability issues of ZnTPP and TPP.

In **Figure R2** below, the four catalytic processes in O-RQP (i.e., photosensitisation, O-RQP-I, O-RQP-II and O-RQP-III) were all supported by experimental evidence, literature, computational calculations and/or comparative studies. We added **Figure R2** as **Supplementary Figure S46** in **Supplementary Section 14** of the supplementary information document. We have also added the following paragraph in the manuscript: “*To increase the readability of the supportive discussion of the O-RQP mechanism above, which is composed of a ¹O₂ photosensitisation process, followed by O-RQP-I (electron transfer reaction between ¹O₂ and TEA), O-RQP-II (activation of RAFT by superoxide O₂^{•-}) and O-RQP-III (deactivation of RAFT) processes. we summarised the chain of evidence from both experimental and computational studies, supporting literature and comparative studies as **Supplementary Figure S46, Supplementary Section 14.***”

Figure R2 also includes a range of comparative studies performed using the chemically inert photocatalyst ZnPC (which is known to be stable in excited states^{6,7}) to once again demonstrate the scientific validity of the proposed O-RQP mechanism.

	Photosensitisation	O-RQP-I	O-RQP-II	O-RQP-III
Experimental Evidence	Supplementary Figure S37 Trapping of $^1\text{O}_2$ by DMAN 	1 Supplementary Figure S36b Proof of the $^1\text{O}_2$ -TEA reaction 	2 $\text{O}_2^{\cdot-}$ captured as the product of $^1\text{O}_2$ -TEA reaction in our preceding work, DOI: 10.1002/anie.201909014 3 With O-RQP-selective PCs, PET-RAFT polymerisation can only proceed under air. Comparative studies show only the proposed O-RQP-II is likely (see below).	1 Temporal control and low $M_w/M_n < 1.1$ were obtained (Figure 2 and 5, main text) 2 NMR, MALDI-TOF and chain extension showed great chain-end fidelity (Supplementary Section 4) 3 0.5 equiv. TEA relative to DTPA sustained all along, proving catalytic cycling
Literature Supporting	Generation of $^1\text{O}_2$ by photoexcited porphyrin and phthalocyanine derivatives via photosensitisation has been widely supported and well documented in literature. See Note 1.	The electron transfer reaction between $^1\text{O}_2$ and a tertiary amine like TEA to produce superoxide is a classic reaction that has been long discovered and well-documented. See Note 2.	See Supplementary Section 10.	The O-RQP-III reaction and similar reactions are commonly seen and well-documented in researches of PET-RAFT via the RQP. See Note 3.
Computational Evidence				
Comparative Study O-RQP-II	 1 mL O_2 injected every 10 min 1 mL O_2 only at beginning Purged with N_2 (sealed) 	 ZnPC/MA/DTPA (without TEA/O_2) ZnPC/MA/DTPA/O_2 (without TEA) ZnPC/MA (without TEA/DTPA/O_2) ZnPC/MA/O_2 (without TEA/DTPA) 	 ZnPC/MA/DTPA/O_2 (without TEA) ZnPC/MA/O_2 (without TEA/DTPA) ZnPC/MA/TEA/O_2 (without DTPA) 	 ZnPC/MA/TEA/O_2 (without DTPA)
	 1 O_2 is a requisite for activating the O-RQP. 2 O_2 has to be injected regularly, to compute with consumption of $^1\text{O}_2$ by the solvent DMSO. 3 Exhaustion or removal of O_2 leads to pause of the reaction. ➡ Catalytic amount of O_2 plays a central catalytic role in O-RQP.	 1 ZnPC is inert for OQP & RQP. 2 ZnPC does not photo-initiate polymerisation. ➡ Without O_2, no polymerisation can happen.	 1 ZnPC is inert for OQP in the presence of O_2 2 ZnPC does not photo-initiate polymerisation with O_2 ➡ Without TEA, O_2 cannot be converted to $\text{O}_2^{\cdot-}$ and activate O-RQP.	 1 Without DTPA (RAFT agent), polymerisation does not occur via O-RQP. ➡ $\text{O}_2^{\cdot-}$ has to interact with DTPA to activate polymerisation via O-RQP.

Figure R2 | Chain of evidence and supporting materials for the O-RQP.

Note 1: Porphyrin and phthalocyanine derivatives are commonly used as $^1\text{O}_2$ photosensitisers because of their significant triplet quantum yields; their ability to generate $^1\text{O}_2$ is well-documented in the literature.⁹⁻¹³

Note 2: The electron transfer reaction between $^1\text{O}_2$ and a tertiary amine such as TEA producing superoxide $\text{O}_2^{\cdot-}$ is a classic reaction which has been studied and well-documented.¹⁴⁻¹⁸

Note 3: The O-RQP-III reaction and similar reactions are commonly seen and well-documented in work pertaining to PET-RAFT via the RQP and other photoredox catalysed polymerisation techniques via the RQP.^{8, 19, 20}

Reviewer #3

3. *I am concerning the possible multiple roles of oxygen in radical polymerization. On one hand, oxygen builds a bridge between the excited state of PC and the ground state of the electron donor, TEA, and the RAFT agent, thus facilitating the SET process to generate radicals, as illustrated in the present work. On the other hand, oxygen is a well-known scavenger to carbon-centred radicals, forming peroxide radical and, as such, retarding the rate of radical polymerization with very high chain transfer coefficient.*

Answer: We thank the reviewer for their careful thinking. The subject of the dual roles of O₂ (O₂ paradox) is indeed interesting, which has been highlighted in our preceding work.²¹

The oxygen tolerance feature of PET-RAFT polymerisation has been substantially reported in our and others' publications, where O₂ is rapidly converted to ¹O₂ upon irradiation and further chemically quenched by the solvent DMSO, forming DMSO₂.²²⁻²⁸ In an inert solvent like CH₃CN (which cannot react with ¹O₂), we had previously observed rapid conversion of vast majority of O₂ into ¹O₂ as demonstrated by the fast conversion of dimethylantracene into the corresponding endoperoxide.²⁷ This photosensitisation process has largely prevented the side effect of O₂ on PET-RAFT polymerisation by eliminating ground state O₂ in the first place. We have also proved in previous reports that, DMSO, as the solvent most used in PET-RAFT polymerisation, served as an excellent ¹O₂ scavenger which rapidly eliminates ¹O₂ by forming DMSO₂.^{22, 27} Indeed, we had previously discovered that, degradation of dimethylantracene was significantly retarded in the presence of pure DMSO upon exposure to ¹O₂ photosensitisation (due to competitive consumption of ¹O₂ by DMSO). Indeed, in our previous reports, there is scarcely any difference between the deoxygenated and the non-deoxygenated polymerisations for PET-RAFT polymerisation with porphyrin-based catalysts.^{24, 27} According to our previous reports²¹⁻²⁸ as well as the MALDI-TOF, NMR and chain extension experiments in this study (**Supplementary Section 4**), the obtained polymer from PET-RAFT polymerisation under air were always in excellent quality comparable to deoxygenated systems because of this oxygen tolerance feature.

Notwithstanding the oxygen tolerance feature of PET-RAFT polymerisation in DMSO, the amount of oxygen needed for O-RQP is at the catalytic level. In other words, by rapid ¹O₂ photosensitisation and subsequent solvent/monomer scavenging, the O₂ content is reduced down to a very low level such that polymerisation is unaffected; however, the remaining ¹O₂ in trivial amount is still highly active and sufficient for sustaining PET-RAFT polymerisation via the O-RQP pathway. For example, as shown in **Figure R2**, bottom row, left figure, green squares, despite the fact that the oxygen content was exponentially reduced due to rapid ¹O₂ photosensitisation and solvent/monomer scavenging, the polymerisation rate was remained constant for the first 10 min, slowed down from 10 min to 25 min and completely ceased only after 25 min. Furthermore, as shown in **Figure R2**, bottom row, left figure, blue squares (RQP), although we had purged the reaction solution with pure N₂ for over 30 min, there is still a small ca. 1-2% jump in monomer conversion within the first 5 min. This is because that the trivial amount of O₂ can mediate O-RQP to a small extent, and we were always able to observe that longer deoxygenation leads to less jump and shorter deoxygenation leads to slightly more obvious proceeding of polymerisation at the early stage.

Overall, as evidenced by these experiments and literature, the compatibility between the oxygen tolerance feature and the O-RQP in PET-RAFT polymerisation lies in the level of O₂ quantity. The oxygen tolerance feature allows high-quality polymerisation by preventing the existence of O₂ in large amounts, whereas efficient O-RQP is allowed due to its requirement for only catalytic amount of O₂, able to “breathe” behind ¹O₂ photosensitisation.

4. *The consumption of oxygen in trithiocarbonate mediated RAFT polymerization in the presence of tertiary amine and without deoxygenation process has been reported (Fu, Q.; et al Polym. Chem. 2017, 8, 1519).*

Answer: We thank the reviewer for mentioning this interesting report. However, in this report,²⁹ Qiao and coworkers discussed about photoexcited RAFT agent oxidising TEA forming the RAFT agent anion radical and then reducing O₂ into superoxide O₂^{•-}. This proposed mechanism is truly plausible in Qiao's conditions. However, there are two critical differences that make Qiao's mechanism highly unlikely in our system.

5. *It is not clear how the molecular oxygen behaves as both a catalyst and a retarder to the reaction. If the latter would be considered, the catalytic ability may turn out to be more effective to compensate the inhibition caused by O₂. Furthermore, it was reported that thiocarbonylthio compounds can be oxidized by the peroxide radical in the RAFT polymerization system (Li, C.; et al J. Polym. Sci.: Part A. Polym. Chem. 2011, 49, 1351 & Barner-Kowollik, C. Polym. Chem. 2010, 1, 634). Although they claimed that there was no noticeable side effects observed stemming from the presence of oxygen, the authors are encouraged to address these two important concerns by conciliating the results of the present work and those in the literature.*

Answer: This comment was addressed in the response to reviewer 3's first question. In short, the compatibility between the oxygen tolerance feature and the O-RQP in PET-RAFT polymerisation lies in the level of O₂ quantity, where the oxygen tolerance feature allows high-quality polymerisation by preventing the existence of O₂ in large amount and efficient O-RQP is allowed due to its requirement for only catalytic amount of O₂.

6. *Minor revision: p.8 Fig. 3a: Is this for GS-GS or ES-GS? The indication in the figure is not consistent with the caption.*

Answer: We thank the reviewer for pointing this out. There is a typo in the caption whereas the figure is correct. The correct caption for **Figure 3a** should be "Derivation of the quantum yield of SET between two ground states". This typo has been corrected.

Supporting references:

1. Ghosh, I.; Shaikh, R. S.; König, B., Sensitization-Initiated Electron Transfer for Photoredox Catalysis. **2017**, *56* (29), 8544-8549.
2. Ghosh, I.; Bardagi, J. I.; König, B., Reply to "Photoredox Catalysis: The Need to Elucidate the Photochemical Mechanism". *Angewandte Chemie International Edition* **2017**, *56* (42), 12822-12824.
3. Coles, M. S.; Quach, G.; Beves, J. E.; Moore, E. G., A Photophysical Study of Sensitization-Initiated Electron Transfer: Insights into the Mechanism of Photoredox Activity. *Angewandte Chemie International Edition* **2020**, *59* (24), 9522-9526.
4. Awwad, N.; Bui, A. T.; Danilov, E. O.; Castellano, F. N., Visible-Light-Initiated Free-Radical Polymerization by Homomolecular Triplet-Triplet Annihilation. *Chem-Us* **2020**.
5. Yeow, J.; Joshi, S.; Chapman, R.; Boyer, C., A Self-Reporting Photocatalyst for Online Fluorescence Monitoring of High Throughput RAFT Polymerization. *Angew Chem Int Ed Engl* **2018**, *57* (32), 10102-10106.
6. Wong, E. W. Y.; Walsby, C. J.; Storr, T.; Leznoff, D. B., Phthalocyanine as a Chemically Inert, Redox-Active Ligand: Structural and Electronic Properties of a Nb(IV)-Oxo Complex Incorporating a Highly Reduced Phthalocyanine(4-) Anion. *Inorg Chem* **2010**, *49* (7), 3343-3350.
7. Sorokin, A. B., Phthalocyanine Metal Complexes in Catalysis. *Chemical Reviews* **2013**, *113* (10), 8152-8191.

8. Corrigan, N.; Xu, J. T.; Boyer, C., A Photoinitiation System for Conventional and Controlled Radical Polymerization at Visible and NIR Wavelengths. *Macromolecules* **2016**, *49* (9), 3274-3285.
9. Rybicka-Jasinska, K.; Shan, W.; Zawada, K.; Kadish, K. M.; Gryko, D., Porphyrins as Photoredox Catalysts: Experimental and Theoretical Studies. *J Am Chem Soc* **2016**, *138* (47), 15451-15458.
10. Alberto, M. E.; De Simone, B. C.; Mazzone, G.; Sicilia, E.; Russo, N., The heavy atom effect on Zn(ii) phthalocyanine derivatives: a theoretical exploration of the photophysical properties. *Phys Chem Chem Phys* **2015**, *17* (36), 23595-601.
11. Josefsen, L. B.; Boyle, R. W., Photodynamic therapy and the development of metal-based photosensitisers. *Met Based Drugs* **2008**, *2008*, 276109.
12. Yeow, J.; Shanmugam, S.; Corrigan, N.; Kuchel, R. P.; Xu, J. T.; Boyer, C., A Polymerization-Induced Self-Assembly Approach to Nanoparticles Loaded with Singlet Oxygen Generators. *Macromolecules* **2016**, *49* (19), 7277-7285.
13. Zhang, X. F.; Xu, H. J., Influence of Halogenation and Aggregation on Photosensitizing Properties of Zinc Phthalocyanine (Znpc). *J Chem Soc Faraday T* **1993**, *89* (18), 3347-3351.
14. Smith, W. F., Kinetic evidence for both quenching and reaction of singlet oxygen with triethylamine in pyridine solution. *Journal of the American Chemical Society* **1972**, *94* (1), 186-190.
15. Hiroshi, T.; Teruo, Y.; Hiroo, T., Intermediates and Mechanism of Photo-Oxygenation Reaction of Triethylamine. *Bulletin of the Chemical Society of Japan* **1973**, *46* (10), 3051-3055.
16. Haugen, C. M.; Bergmark, W. R.; Whitten, D. G., Singlet oxygen mediated fragmentation of amino alcohols, 1,2-diamines, and amino ketones. *Journal of the American Chemical Society* **1992**, *114* (26), 10293-10297.
17. Bernstein, R.; Foote, C. S., Singlet Oxygen Involvement in the Photochemical Reaction of C60 and Amines. Synthesis of an Alkyne-Containing Fullerene. *The Journal of Physical Chemistry A* **1999**, *103* (36), 7244-7247.
18. Cocquet, G.; Rool, P.; Ferroud, C., Photosensitized oxidation, by single-electron transfer, of catharanthine and vindoline: a highly regio- and diastereoselective photocyanation reaction. *Journal of the Chemical Society, Perkin Transactions 1* **2000**, (14), 2277-2281.
19. Liu, X. D.; Zhang, L. F.; Cheng, Z. P.; Zhu, X. L., Metal-free photoinduced electron transfer-atom transfer radical polymerization (PET-ATRP) via a visible light organic photocatalyst. *Polym Chem-Uk* **2016**, *7* (3), 689-700.
20. Xu, J. T.; Shanmugam, S.; Duong, H. T.; Boyer, C., Organo-photocatalysts for photoinduced electron transfer-reversible addition-fragmentation chain transfer (PET-RAFT) polymerization. *Polym Chem-Uk* **2015**, *6* (31), 5615-5624.
21. Zhang, L. W.; Wu, C. Y.; Jung, K.; Ng, Y. H.; Boyer, C., An Oxygen Paradox: Catalytic Use of Oxygen in Radical Photopolymerization. *Angew Chem Int Edit* **2019**, *58* (47), 16811-16814.
22. Corrigan, N.; Rosli, D.; Jones, J. W. J.; Xu, J. T.; Boyer, C., Oxygen Tolerance in Living Radical Polymerization: Investigation of Mechanism and Implementation in Continuous Flow Polymerization. *Macromolecules* **2016**, *49* (18), 6779-6789.
23. Ng, G.; Yeow, J.; Xu, J. T.; Boyer, C., Application of oxygen tolerant PET-RAFT to polymerization-induced self-assembly. *Polym Chem-Uk* **2017**, *8* (18), 2841-2851.
24. Shanmugam, S.; Xu, J.; Boyer, C., Exploiting Metalloporphyrins for Selective Living Radical Polymerization Tunable over Visible Wavelengths. *J Am Chem Soc* **2015**, *137* (28), 9174-85.
25. Shanmugam, S.; Xu, J. T.; Boyer, C., Aqueous RAFT Photopolymerization with Oxygen Tolerance. *Macromolecules* **2016**, *49* (24), 9345-9357.
26. Wu, C.; Corrigan, N.; Lim, C. H.; Jung, K.; Zhu, J.; Miyake, G.; Xu, J.; Boyer, C., Guiding the Design of Organic Photocatalyst for PET-RAFT Polymerization: Halogenated Xanthene Dyes. *Macromolecules* **2019**, *52* (1), 236-248.
27. Wu, C.; Shanmugam, S.; Xu, J.; Zhu, J.; Boyer, C., Chlorophyll a crude extract: efficient photo-degradable photocatalyst for PET-RAFT polymerization. *Chem Commun (Camb)* **2017**, *53* (93), 12560-12563.
28. Wu, C.; Chen, H.; Corrigan, N.; Jung, K.; Kan, X.; Li, Z.; Liu, W.; Xu, J.; Boyer, C., Computer-Guided Discovery of a pH-Responsive Organic Photocatalyst and Application for pH and Light Dual-Gated Polymerization. *J Am Chem Soc* **2019**, *141* (20), 8207-8220.
29. Fu, Q.; Xie, K.; McKenzie, T. G.; Qiao, G. G., Trithiocarbonates as intrinsic photoredox catalysts and RAFT agents for oxygen tolerant controlled radical polymerization. *Polym Chem-Uk* **2017**, *8* (9), 1519-1526.

REVIEWERS' COMMENTS

Reviewer #1 (Remarks to the Author):

The authors have done an excellent job at addressing the concerns of the reviewers. This manuscript is now suitable for publication.

Reviewer #2 (Remarks to the Author):

This paper has been well revised and the additional comment almost convinced me. Although I still think this is a little bit premature, the new plausible mechanism seems very interesting and, in fact, the authors have succeeded in controlling the polymerization using this catalyst as I commented. And the mechanism will become clearer with further investigation in the future. Therefore, I agree with publication of this article as a communication in Nat Commun.

Reviewer #3 (Remarks to the Author):

Comments on the revised manuscript NCOMMS-20-22632A "Unravelling an Oxygen-mediated..." by Cyrille Boyer:

The authors give convincing explanation to my concerns about the dual role of oxygen in radical polymerization by the statement "In short, the compatibility between the oxygen tolerance feature and the O-RQP RAFT polymerization lies in the level of O₂ quantity, ..." I therefore suggest that the author clarify this point by defining the present RAFT system in terms of oxygen content at least in the abstract and in the conclusion. Some discussions in the text in comparison with the mentioned refs. (J. Polym. Sci. Part A: Polym Chem. 2011, 49, 1351 and Polym. Chem (UK) 2010, 1, 634) would also be helpful for better understanding by the readers.

It seems that the authors give an uncompleted response for the Question 4, i.e. what are the two critical differences between your system and Qiao's mechanism ?

I strongly recommend the publication of the manuscript after addressing the above two points.

Response to Reviews (Round 2):

Reviewer #3

1. *The authors give convincing explanation to my concerns about the dual role of oxygen in radical polymerization by the statement "In short, the compatibility between the oxygen tolerance feature and the O-RQP RAFT polymerization lies in the level of O₂ quantity, ..." I therefore suggest that the author clarify this point by defining the present RAFT system in terms of oxygen content at least in the abstract and in the conclusion. Some discussions in the text in comparison with the mentioned refs. (J. Polym. Sci. Part A: Polym Chem. 2011, 49, 1351 and Polym. Chem (UK) 2010, 1, 634) would also be helpful for better understanding by the readers.*

Response: We agree with the reviewer and this point indeed needs to be both clarified in the text and mentioned in the abstract and the concluding paragraph.

Hence, we added the following paragraph to the main text where appropriate with comparison of the two mentioned references^{1,2} with the present study as below:

"As suggested by our previous contribution, the dual role of O₂ as a radical scavenger and a catalytic species, is a "paradox".³ Specifically, the propagating carbon radical in a polymerisation is subject to oxidation by ground state O₂ leading to the formation of a peroxy radical¹ and degradation of the dormant polymer chain may also occur in the presence of ground state O₂.² However, unlike conventional, thermally initiated RAFT polymerisations, the PET-RAFT systems possesses excellent oxygen tolerance as reported in previous publications, where O₂ is rapidly converted to ¹O₂ upon irradiation and chemically quenched by the solvent DMSO, forming DMSO₂.⁴⁻¹⁰ In this context, the interplay between the oxygen tolerance feature and the catalytic role of O₂ in the O-RQP mechanism lies in the amount of O₂; the oxygen tolerance feature facilitates high-quality polymerisation by lowering the O₂ concentration to the catalytic amounts required for efficient O-RQP."

Also, we added the following sentence in the abstract to reflect this point:

"where the oxygen tolerance of PET-RAFT allows high-quality polymerisation by preventing the existence of O₂ in large amounts and efficient O-RQP is permitted due to its requirement for only catalytic amounts of O₂."

In the conclusion, we added:

"which eliminates sufficient O₂ to the point where radical polymerisation proceeds unencumbered whilst the remaining catalytic amounts of O₂ serve as a cocatalyst to provide facile polymerisation."

Response:

2. *It seems that the authors give an uncompleted response for the Question 4, i.e. what are the two critical differences between your system and Qiao's mechanism?*

Response: We apologise for the incomplete response. The "two critical differences between your (our) system and Qiao's mechanism" are:

First and foremost, throughout our work, light sources > 590 nm up to 780 nm were implemented; at these extended wavelengths the RAFT agent cannot be photoexcited (there is no photoexcitation of the RAFT agent in our mechanism), which makes Qiao's mechanism irrelevant to our study (Qiao's mechanism relies on photoexcitation of the RAFT agent to activate the photocatalytic system; Qiao's mechanism is only possible with a photoexcited RAFT agent). Second, due to the highly efficient photosensitiser present in our systems, O₂ was rapidly converted to ¹O₂ and is unlikely to react with the RAFT anion radical, which further supports the unlikelihood of Qiao's mechanism.

- 1 Li, C., He, J., Zhou, Y., Gu, Y. & Yang, Y. Radical-induced oxidation of RAFT agents—A kinetic study. *Journal of Polymer Science Part A: Polymer Chemistry* **49**, 1351-1360, doi:<https://doi.org/10.1002/pola.24554> (2011).
- 2 Gruending, T., Weidner, S., Falkenhagen, J. & Barner-Kowollik, C. Mass spectrometry in polymer chemistry: a state-of-the-art up-date. *Polym Chem-Uk* **1**, 599-617, doi:10.1039/B9PY00347A (2010).
- 3 Zhang, L. W., Wu, C. Y., Jung, K., Ng, Y. H. & Boyer, C. An Oxygen Paradox: Catalytic Use of Oxygen in Radical Photopolymerization. *Angew Chem Int Edit* **58**, 16811-16814, doi:10.1002/anie.201909014 (2019).
- 4 Corrigan, N., Rosli, D., Jones, J. W. J., Xu, J. T. & Boyer, C. Oxygen Tolerance in Living Radical Polymerization: Investigation of Mechanism and Implementation in Continuous Flow Polymerization. *Macromolecules* **49**, 6779-6789, doi:10.1021/acs.macromol.6b01306 (2016).
- 5 Ng, G., Yeow, J., Xu, J. T. & Boyer, C. Application of oxygen tolerant PET-RAFT to polymerization-induced self-assembly. *Polym Chem-Uk* **8**, 2841-2851, doi:10.1039/c7py00442g (2017).
- 6 Shanmugam, S., Xu, J. & Boyer, C. Exploiting Metalloporphyrins for Selective Living Radical Polymerization Tunable over Visible Wavelengths. *J Am Chem Soc* **137**, 9174-9185, doi:10.1021/jacs.5b05274 (2015).
- 7 Shanmugam, S., Xu, J. T. & Boyer, C. Aqueous RAFT Photopolymerization with Oxygen Tolerance. *Macromolecules* **49**, 9345-9357, doi:10.1021/acs.macromol.6b02060 (2016).
- 8 Wu, C. *et al.* Guiding the Design of Organic Photocatalyst for PET-RAFT Polymerization: Halogenated Xanthene Dyes. *Macromolecules* **52**, 236-248, doi:10.1021/acs.macromol.8b02517 (2019).
- 9 Wu, C., Shanmugam, S., Xu, J., Zhu, J. & Boyer, C. Chlorophyll a crude extract: efficient photo-degradable photocatalyst for PET-RAFT polymerization. *Chem Commun (Camb)* **53**, 12560-12563, doi:10.1039/c7cc07663k (2017).
- 10 Wu, C. *et al.* Computer-Guided Discovery of a pH-Responsive Organic Photocatalyst and Application for pH and Light Dual-Gated Polymerization. *J Am Chem Soc* **141**, 8207-8220, doi:10.1021/jacs.9b01096 (2019).